# How lamina-associated polypeptide 1 (LAP1) activates Torsin

Brian A Sosa[1], F Esra Demircioglu[1], James Z Chen[1], Jessica Ingram[1,2], Hidde L Ploegh[1,2], Thomas U Schwartz[1]*

[1]Department of Biology, Massachusetts Institute of Technology, Cambridge, United States; [2]Whitehead Institute for Biomedical Research, Cambridge, United States

**Abstract** Lamina-associated polypeptide 1 (LAP1) resides at the nuclear envelope and interacts with Torsins, poorly understood endoplasmic reticulum (ER)-localized AAA+ ATPases, through a conserved, perinuclear domain. We determined the crystal structure of the perinuclear domain of human LAP1. LAP1 possesses an atypical AAA+ fold. While LAP1 lacks canonical nucleotide binding motifs, its strictly conserved arginine 563 is positioned exactly where the arginine finger of canonical AAA+ ATPases is found. Based on modeling and electron microscopic analysis, we propose that LAP1 targets Torsin to the nuclear envelope by forming an alternating, heterohexameric (LAP1-Torsin)$_3$ ring, in which LAP1 acts as the Torsin activator. The experimental data show that mutation of arginine 563 in LAP1 reduces its ability to stimulate TorsinA ATPase hydrolysis. This knowledge may help scientists understand the etiology of DYT1 primary dystonia, a movement disorder caused by a single glutamate deletion in TorsinA.

*For correspondence: tus@mit.edu

## Introduction

Lamina-Associated Polypeptide 1 (LAP1) and Luminal domain-Like LAP1 (LULL1) are type II integral membrane proteins with ~30 kDa luminal domains that share 62% identity. LAP1 localizes to the nuclear envelope (NE) via its lamin-interacting domain, whereas LULL1 is found throughout the endoplasmic reticulum (ER) (*Senior and Gerace, 1988*; *Goodchild and Dauer, 2005*). LAP1 and LULL1 associate with Torsins (*Goodchild and Dauer, 2005*; *Naismith et al., 2009*; *Zhao et al., 2013*), which are ER-resident members of the ATPases Associated with a variety of cellular Activities (AAA+ ATPases) superfamily (*Iyer et al., 2004*; *Hanson and Whiteheart, 2005*; *Erzberger and Berger, 2006*). AAA+ ATPases use ATP hydrolysis to undergo conformational changes and to exert mechanical force on a substrate. AAA+ ATPases are involved in many processes, including vesicle fusion and scission, protein folding/unfolding, complex assembly/disassembly, protein transport, and nucleic acid remodeling. Detailed mechanistic knowledge exists for a number of AAA+ ATPases, but not for Torsins. Torsins are found in all animals, but not in single-cell organisms and plants. Deletion of a single glutamate in TorsinA (humans have four Torsin orthologs: TorsinA, TorsinB, Torsin2A, and Torsin3A) at position 302/303 causes the movement disorder DYT1 primary dystonia (*Ozelius et al., 1997*; *Breakefield et al., 2008*). Mammalian TorsinA has been assigned to a variety of possible functions including nuclear envelope (NE) organization, synaptic vesicle transport and turnover, operation of the secretory pathway, protein degradation, cytoskeletal organization and transport via NE budding (*Vander Heyden et al., 2009*; *Granata and Warner, 2010*; *Jokhi et al., 2013*). While LAP1 and LULL1 were first considered as Torsin substrates (*Naismith et al., 2009*), they are now being proposed as possible activators of Torsin (*Zhao et al., 2013*).

To better understand the function of LAP1 in relation to Torsin, we set out to determine its structure. Our data suggest that LAP1 and LULL1 can both form heterohexameric ring assemblies with

**eLife digest** Cells are filled with activity—proteins must be folded into intricate shapes and unfolded, complex molecules must be built and taken apart—and all this activity requires energy. Cells use a molecule called ATP as a source of energy, with the energy being released when enzymes called ATPases remove a phosphate group from the ATP molecule.

There are many different ATPases, and when one of them does not work properly, the consequences can be severe. A single mutation in the gene for an ATPase called TorsinA, for example, can lead to a painful and severely disabling disorder called primary dystonia. However, scientists have yet to discover what the TorsinA enzyme does in cells and how mutated TorsinA causes primary dystonia.

It is known that the TorsinA enzyme is found in the endoplasmic reticulum—a region of the cell where, among other things, proteins are folded and unfolded—and that it interacts with two membrane proteins, LAP1 and LULL1. Also, TorsinA is a member of the AAA+ family of ATPases (AAA+ is short for ATPases Associated with diverse cellular Activities), and these enzymes tend to combine with each other to form functional ring-shaped arrays.

Now Sosa et al. have used a combination of X-ray crystallography, electron microscopy, computer modeling, and biochemistry to study the interactions between TorsinA and the two membrane proteins. This work revealed that, somewhat surprisingly, certain parts of the membrane proteins have structures that are similar to those of AAA+ ATPases. This allows the TorsinA enzymes and the membrane proteins to combine to form rings that contain three enzymes and three LAP1 proteins or three LULL1 proteins.

Within these rings, the LAP1 and LULL1 proteins activate the TorsinA enzyme by supplying an amino acid called arginine to the neighboring TorsinA molecule. The amino acid is supplied to a site on the enzyme that can bind nucleotides (which are the building blocks of DNA and RNA); the LAP1 and LULL1 proteins themselves cannot bind nucleotides.

The work of Sosa et al. may help scientists to better understand how TorsinA causes primary dystonia. One of the important next steps is to work out what molecules TorsinA acts on.

Torsin, whereby Torsin is activated through an arginine finger at amino acid (aa) R563 of LAP1 (R449 of LULL1). Thus LAP1 and LULL1 each have dual roles, namely the targeting and activation of Torsins.

## Results

The conserved luminal domain of LAP1 (aa356–583) was recombinantly expressed, purified, and set up for crystallization. Because the protein failed to yield crystals alone, we produced a camelid single domain (VHH) antibody fragment, VHH-BS1, against LAP1 as a crystallization chaperone. The complex of VHH-BS1 and LAP1 yielded well-diffracting crystals (*Table 1*). The structure was solved using molecular replacement with VHH as the search model (for details, see 'Materials and methods'). In the asymmetric unit we find two LAP1 molecules related by a non-crystallographic symmetry axis, each identically bound by one VHH. The final structure is completely built, except for four N-terminal residues in LAP1, which seem disordered in the crystal (six N-terminal residues in the second LAP1 copy).

The LAP1 structure clearly has an AAA+ like fold (*Figure 1*). AAA+ proteins are a functionally diverse group within the vast family of 'P-loop'-type NTP-binding proteins (*Iyer et al., 2004*). Typically, AAA+ proteins are characterized by strongly conserved sequence motifs involved in nucleotide recognition. Subcategories within AAA+ proteins are based on distinct secondary structure topologies. The D2 domain of the two-ring AAA+ ATPase ClpB superimposes on LAP1 with an rmsd of 3.28 Å over 111 Cα positions (*Figure 1*; *Zeymer et al., 2014*). ClpB belongs to the additional strand conserved E (ASCE) family (*Iyer et al., 2004*). Like ASCE-type AAA+ proteins, such as ClpA, ClpB, and p97, LAP1 also has a central, five-stranded parallel β-sheet, surrounded on both sides by 10 helices in total (*Figure 1A,B*). The secondary structure topology is α(−1)α(0)β1α(1)β2-α(2a)α(2b)β3α(3a)α(3b)β4α(4)β5 α(5)α(6), using the AAA+ nomenclature (*Erzberger and Berger, 2006*). In contrast to canonical ASCE-type ATPases (*Figure 1C*), LAP1 lacks a C-terminal domain. Instead the C terminus is attached to helix α1 via a disulfide bond between the highly conserved residues C424 and C582 (*Figure 1—figure supplement 1*). In a AAA+ ATPase, the nucleotide binding pocket is the most sequence-conserved

**Table 1.** X-ray data collection and refinement statistics

| Protein | Human LAP1$_{356-583}$-VHH-BS1 complex |
|---|---|
| PDB ID | 4TVS |
| **Data collection** | |
| Space group | P2$_1$ |
| a, b, c (Å) | 69.79, 74.02, 85.43 |
| α, ß, γ (°) | 90, 108.8, 90 |
| Wavelength (Å) | 1.2548 |
| Resolution range (Å)* | 66.1–1.60 (1.66–1.60) |
| Total reflections | 630,312 (46,516) |
| Unique reflections | 105,976 (10,195) |
| Completeness (%) | 97.6 (94.3) |
| Redundancy | 5.9 (4.6) |
| Rsym (%) | 9.1 (141.2) |
| Rp.i.m. (%) | 4.0 (68.6) |
| I/σ | 12.5 (1.1) |
| CC$_{1/2}$ (%) | 99.8 (54.7) |
| **Refinement** | |
| Resolution range (Å) | 66.1–1.60 |
| R$_{work}$ (%) | 18.1 |
| R$_{free}$ (%) | 22.9 |
| Coordinate error (Å)† | 0.24 |
| Number of reflections | |
| Total | 105,565 |
| R$_{free}$ reflections | 2810 |
| Number of non-hydrogen atoms | |
| Protein | 5520 |
| Ligands | 29 |
| Water | 612 |
| R.m.s. deviations | |
| Bond lengths (Å) | 0.019 |
| Bond angles (°) | 1.66 |
| B-factors (Å²) | |
| Protein | 39.3 |
| Ligands | 56.4 |
| Water | 47.8 |
| Ramachandran (%)‡ | |
| Favored (%) | 98.3 |
| Outlier (%) | 0.0 |
| Clashscore | 5.65 |
| MolProbity score‡ | 1.29 |
| MolProbity percentile‡ | 96th |

*Numbers in brackets refer to the highest resolution shell (10% of all reflections).

*Table 1. Continued on next page*

region, defined by the essential Walker A and B motifs. LAP1 lacks a recognizable Walker A region, typically located between β1 and α1 (*Figure 2A,B*, *Figure 1—figure supplement 1*). Instead, helix α1 is N-terminally extended compared to AAA+ ATPases, thereby sterically blocking the nucleotide-binding pocket. The Walker B motif with the canonical sequence hhhhDE (h, hydrophobic), is instead hhhhHR in human LAP1, and therefore cannot interact with nucleotide. Finally, the conserved disulfide bridge would also interfere with nucleotide binding (*Figure 2B*, *Figure 1—figure supplement 1*).

Having established that LAP1 represents a nucleotide-free AAA+ domain, we asked how this might inform us about LAP1's interaction with Torsins, which can be modeled with great confidence as AAA+ ATPases (*Zhu et al., 2010*). The four canonical nucleotide-sensing elements, Walker A, Walker B, Sensor 1, and Sensor 2 (*Erzberger and Berger, 2006*), are well conserved and immediately recognizable in TorsinA when compared to ClpB-D2 (*Figure 2C,D*, *Figure 2—figure supplement 1*; *Kock et al., 2006*; *Zhu et al., 2010*). AAA+ ATPases are often activated by an arginine residue ('arginine finger') in the neighboring protomer in the hexameric ring assembly, such that this arginine is positioned to reach the phosphate binding site (*Wendler et al., 2012*). While we do not find a conserved arginine in TorsinA (*Figure 2—figure supplement 1*) at the expected position at the end of helix α5, in LAP1 this residue is a strictly conserved arginine (R563). Consequently, modeling and phylogenetic analysis strongly suggest that LAP1 and TorsinA might form a heterohexameric ring (*Figure 3A,B*) with three active sites in which TorsinA and its partner subunits alternate to form the ring. This hypothesis is further strengthened by the recent observation that LAP1 and LULL1 activate TorsinA in vitro (*Zhao et al., 2013*).

To experimentally confirm assembly of LAP1 and TorsinA into a heterohexameric ring, we co-expressed and co-purified TorsinA(E171Q) in complex with either LAP1 and LULL1 via nickel-affinity and size-exclusion chromatography. We obtained complexes with TorsinA(E171Q):LAP1 and TorsinA(E171Q):LULL1 1:1 stoichiometry (*Figure 3—figure supplement 1*). The E171Q mutation traps TorsinA in the ATP-bound form, which helps to stabilize the interaction with LAP1 and LULL1, respectively (*Goodchild and Dauer, 2005*). For negative-stain electron microscopic analysis, the nickel eluate was used directly. On individual micrographs, rings of expected size

*Table 1. Continued*

†Maximum likelihood based (as determined by PHENIX; ***Adams et al., 2010***).

‡As determined by MolProbity (***Chen et al., 2010***).

were observed for the two-complex preparation, but not for individually purified LAP1(356–583) or LULL1(322–570) (***Figure 3—figure supplement 2***). TorsinA(E171Q) is insoluble without binding partner, therefore the observed rings should not be homomeric (***Figure 1—figure supplement 3***). The best micrographs, obtained with the TorsinA(E171Q):LULL1 preparation, were further processed. Class averaging of 808 TorsinA(E171Q):LULL1 particles yielded multiple classes showing ring assembly (***Figure 3C***). The rings have a diameter of ~120 Å, in good agreement with typical hexameric AAA+ ATPase assemblies.

We then tested the ATPase activity of TorsinA in the context of LAP1 and LULL1. In comparison to wild-type TorsinA:LAP1, the mutants TorsinA:LAP1(R563A) and TorsinA:LULL1(R449A) showed substantial reduction in ATPase activity, while both TorsinA(E171Q):LAP1 and TorsinA(E171Q):LULL1 are essentially inactive (***Figure 4***). We note that the ATP hydrolysis rates are very slow and that the effect of the arginine finger mutation is not as drastic as seen with other AAA+ ATPases. We speculate that the ATP hydrolysis rate in the presence of substrate is likely higher, and that the effect of the arginine finger is likely more pronounced in a more physiological context.

In summary, we conclude that LAP1 and LULL1 both activate Torsins by providing the arginine finger in a heterohexameric ring assembly in which LAP1 and LULL1 alternate with Torsin. Since LAP1 and LULL1 contain transmembrane helices immediately N-terminal to their luminal, nucleotide-free AAA+ Activator domain, they bring Torsin into close proximity to the membrane (***Figure 5***).

## Discussion

Here we provide structural evidence to explain how LAP1 and LULL1 function as activating proteins for Torsin. This confirms recent work which biochemically showed that LAP1 and LULL1 activate TorsinA (***Zhao et al., 2013***). A heterohexameric ring assembly for AAA+ ATPases is unusual, but not unprecedented (***Gribun et al., 2008***; ***Saffian et al., 2012***). However, to our knowledge this is the first

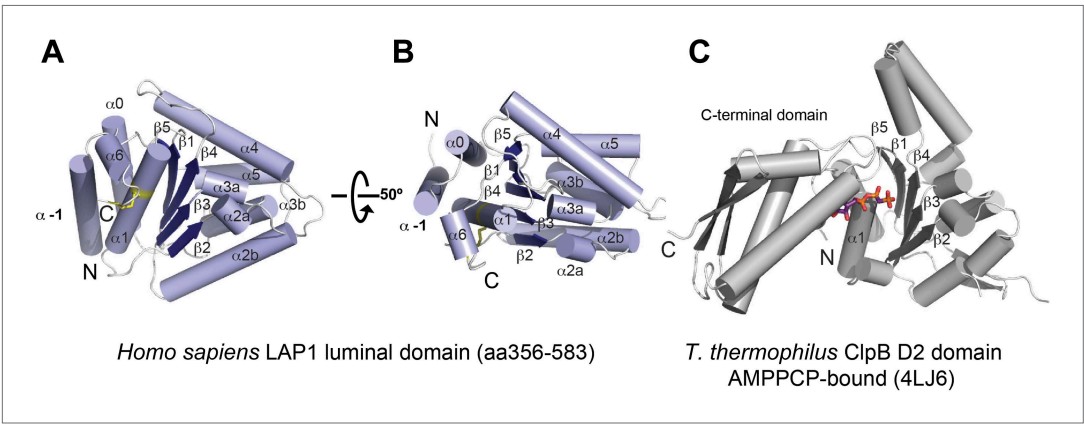

*Homo sapiens* LAP1 luminal domain (aa356-583)

*T. thermophilus* ClpB D2 domain AMPPCP-bound (4LJ6)

**Figure 1**. Crystal structure of human LAP1. (**A**) Crystal structure of the luminal domain of LAP1 from *Homo sapiens*. The helices are numbered according to the nomenclature used for AAA+ proteins (***Erzberger and Berger, 2006***). A disulfide bridge (yellow) attaches the C terminus to helix α1. (**B**) Same as (**A**), but rotated around the x-axis by 50°. (**C**) Crystal structure of the AMPPCP-bound ClpB-D2 domain from *Thermus thermophilus* (4LJ6; ***Zeymer et al., 2014***) in the same orientation as LAP1 in (**A**) revealing the striking topological similarity. In comparison to ClpB, LAP1 does not bind a nucleotide and has no C-terminal domain.

The following figure supplements are available for figure 1:

**Figure supplement 1**. Phylogenetic analysis of LAP1 homologs.

**Figure supplement 2**. Representative 2Fo-Fc electron density of the final model.

**Figure supplement 3**. VHH-BS1 competes with TorsinA for LAP1 binding.

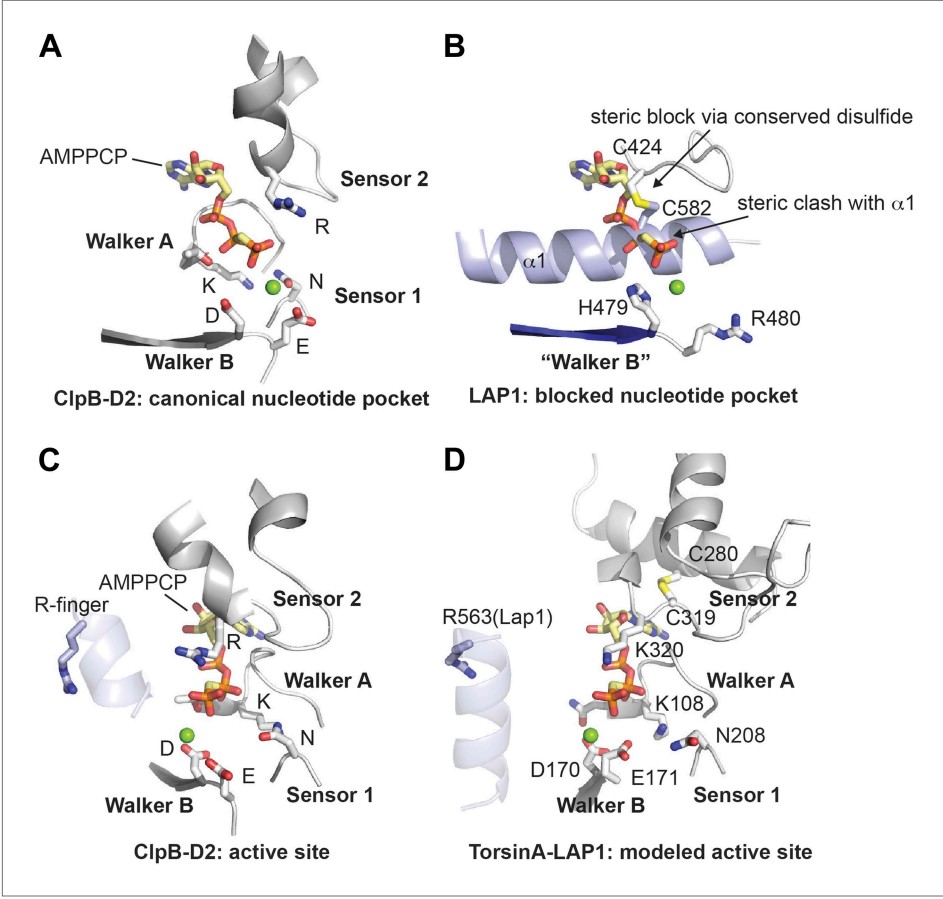

**Figure 2**. Nucleotide binding site. (**A** and **B**) Comparison of the nucleotide binding pocket in ClpB-D2 (**A**) with the equivalent region in LAP1 (**B**). The nucleotide sensing elements of canonical ATPases are indicated. The nucleotide in (**B**) is modeled to illustrate how LAP1 blocks binding of it. (**C** and **D**) To activate ATP hydrolysis, an arginine residue (R-finger) from the neighboring protomer in the typical hexameric ring assembly is necessary. For ClpB-D2, this is R747 (light blue) (**C**). Modeled as a heterohexameric LAP1-TorsinA assembly, the strictly conserved R563 in LAP1 is positioned as an R-finger to point into the nucleotide-binding pocket of a neighboring TorsinA protomer (**D**). Residues important for nucleotide interaction are labeled in the TorsinA model.

The following figure supplement is available for figure 2:

**Figure supplement 1**. Phylogenetic analysis of Torsin.

description of a heterohexameric assembly where a canonical AAA+ domain alternates with a nucleotide-free AAA+ like domain, which we now call AAA+ Activator domain. It is striking that LAP1 and LULL1 lack all canonical nucleotide-binding elements. This feature has undoubtedly complicated their detection by sequence-based methods. Using the LAP1 structure, we performed a reverse hidden Markov model search to possibly identify structurally similar proteins in various organisms using Backphyre (***Kelley and Sternberg, 2009***). Interestingly, the top hits that we find besides LAP1 homologs are the Torsins. This indicates that both proteins are possibly derived from a common ancestor. In Drosophila as well as in Caenorhabditis, we find LAP1 homologs that have so far escaped detection (***Figure 1—figure supplement 1***). Gratifyingly, for all species where so far an orphaned Torsin has been found, we now also find a LAP1/LULL1 homolog (for many species, it is difficult to distinguish between LAP1 and LULL1 based on sequence alone). This lends further support to our model suggesting that a heterohexameric LAP1-Torsin ring is the functionally relevant protein assembly.

Our structure was obtained using a VHH raised against LAP1. We can only speculate as to why VHH-BS1 helped in crystallizing the protein. Perhaps VHH-BS1 fortuitously generates packing contacts

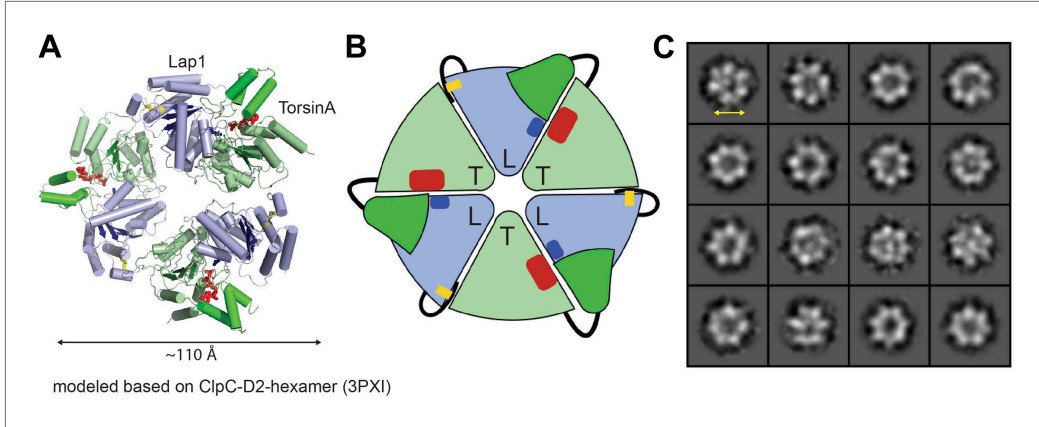

**Figure 3**. Heterohexameric ring assembly. (**A**) Heterohexameric model of alternating LAP1 (light blue) and TorsinA (green hues), based on the hexameric ring of ClpC-D2 domains (3PXI; *Wang et al., 2011*). The C-terminal domain of Torsin is colored bright green. An ATP molecule (red) is modeled into the TorsinA nucleotide binding pocket. The conserved arginine finger in LAP1 is in blue. The disulfide bridge within LAP1 is in yellow. (**B**) Schematic diagram of (**A**), same coloring scheme. (**C**) A montage of representative particle classes of the TorsinA(E171Q):LULL1 complex obtained from negatively stained particles are shown. The particles are consistent with a toroidal hexameric conformation as shown in (**A** and **B**). The particle diameter is approximately 120 Å (double arrow).

The following figure supplements are available for figure 3:

**Figure supplement 1**. Analytical gel filtration of TorsinA(E171Q):LAP1 and TorsinA(E171Q):LULL1 complexes.

**Figure supplement 2**. Negative-stain micrographs.

important for crystal formation. We note that VHH-BS1 binds to the area immediately surrounding the arginine finger of LAP1, thereby interfering with TorsinA binding (*Figure 1—figure supplement 3*). Inclusion of VHH-BS1 might prevent cryptic and poorly ordered hexamerization of LAP1 at high protein concentrations and therefore favor a more ordered conformation conducive to crystallization. Interestingly, VHH-BS1 can compete with LAP1 for TorsinA binding in vitro (*Figure 1—figure supplement 3*). When expressed intracellularly, it might be a useful tool to further characterize TorsinA-LAP1 function in vivo.

What are the substrate(s) of Torsin? Torsin acts in close proximity to the NE or the ER membrane, depending on whether it interacts with membrane-bound LAP1 or LULL1, respectively. Membrane proximity is further ensured by the N-terminal hydrophobic region within TorsinA (*Vander Heyden et al., 2011*). Using an ATP-trapped mutant, it was shown that TorsinA is involved in the exit of large ribonucleoprotein (mRNP) granules from the nucleus through a pathway akin to the nuclear egress of herpes-type viruses (*Rose and Schlieker, 2012*; *Speese et al., 2012*; *Jokhi et al., 2013*). A substrate-trap mutant of TorsinA localized to the neck of mRNP-filled vesicles that bud from the inner nuclear membrane (INM) into the perinuclear space (*Jokhi et al., 2013*), consistent with our model of LAP1-mediated activation of TorsinA close to the INM. We therefore speculate that the so far elusive Torsin substrates include proteins involved in membrane scission. This is akin to the action of the AAA+ ATPase Vps4 involved in the scission of narrow membrane necks mediated by ESCRT-III (endosomal sorting complexes required for transport) components (*Hill and Babst, 2012*; *McCullough et al., 2013*), but with an important difference: while Vps4 acts on the inside of the necks, Torsins presumably act on the outside. We further speculate that under more physiological conditions, that is, properly complexed with LAP1 or LULL1 and engaged with substrate, the ATPase activity of Torsin might be substantially higher than that reported under the in vitro conditions tested so far.

Apart from the identification of Torsin substrate(s), understanding the catalytic mechanism of the Torsin-LAP1/LULL1 machinery is equally important. Because of the unusual heterohexameric architecture with only three active sites, we expect substantial differences between this system and those that are better understood. One important aspect will be to unravel the role of the highly conserved

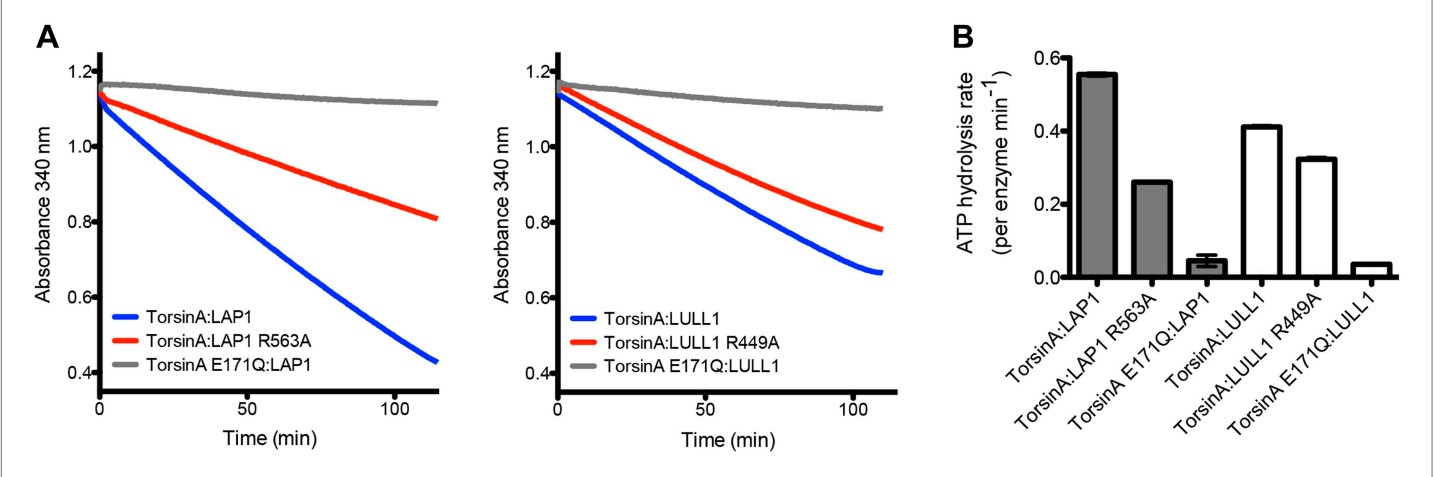

**Figure 4**. ATPase assay. (**A**) TorsinA:LAP1 and TorsinA:LULL1 complexes were tested for ATPase activity in a coupled ADP/NADH assay, where the oxidation of NADH is monitored spectrometrically (**Norby, 1988**). Shown are the oxidation rate plots, from which the ATP hydrolysis rates are calculated using linear regression. Reactions were performed in triplicate. (**B**) Bar graph representation of ATPase assay results shown in (**A**).

cysteines, both in LAP1/LULL1 and in Torsin. While both proteins have conserved disulfides, these do not account for all cysteines and suggest a possible redox mechanism, as has been discussed previously (**Zhu et al., 2010**, **2008**). We note that in our model the most conserved cysteine C496 in LAP1 cannot engage in an intramolecular disulfide bridge, but is close to the nucleotide-binding site of the neighboring TorsinA. This remarkable conservation might indicate a role in a redox mechanism that is coupled to the catalytic cycle of Torsin-LAP1/LULL1. The lumen of the endoplasmic reticulum is a site rich in oxidoreductases that assist in protein folding and assembly. The catalytic function(s) of the Torsin-LAP1/LULL1 assemblies may have co-opted part of this machinery.

## Materials and methods

### Plasmids, protein expression, and purification

Recombinant proteins were expressed in *Escherichia coli*. Human TorsinA (aa51–322), the luminal domain of human LAP1 (aa356–583), and the luminal domain of human LULL1 (aa233–470) were expressed from a modified ampicillin resistant pETDuet-1 (EMD Millipore, Billerica, MA) vector as N-terminally 6×His-7×Arg-tagged fusion proteins. A cleavage site for human rhinovirus 3C protease was inserted after the 7×Arg tag. For co-expression of TorsinA with LAP1 or LULL1, cells were co-transformed with the TorsinA vector described above and a second, modified kanamycin resistant pETDuet-1 vector containing untagged LAP1 or LULL1. VHH-BS1 was expressed as a C-terminally 6×His-tagged fusion protein from a kanamycin resistant pET-28b(+) (EMD Millipore) vector. Mutations were introduced by site-directed mutagenesis.

All proteins were expressed in LOBSTR(DE3) RIL strains (Kerafast, Boston, MA) (**Andersen et al., 2013**). Bacterial cultures were grown at 30°C to an optical density (OD600) of 0.6, shifted to 18°C for 30 min, and induced overnight at 18°C with 0.2 mM IPTG. LAP1 and VHH-BS1 expressing cells were resuspended in lysis buffer A (50 mM potassium phosphate pH 8.0, 400 mM NaCl, 40 mM imidazole) and lysed. The lysate was supplemented with 1 U/ml Benzonase (Sigma-Aldrich, St. Louis, MO) and 1 mM PMSF, cleared by centrifugation, and loaded onto a Ni-affinity resin. After washing with lysis buffer, bound protein was eluted with elution buffer (10 mM potassium phosphate pH 8.0, 150 mM NaCl, 250 mM imidazole). For LAP1, the eluted protein was purified by cation-exchange chromatography against a gradient of 0.150–2 M NaCl with 10 mM potassium phosphate pH 8.0, followed by dialysis into cleavage buffer (10 mM potassium phosphate pH 8.0, 150 mM NaCl) and tag removal with 3C protease followed by another round of cation-exchange chromatography to remove tag and protease. The flow-through from the cation-exchange chromatography was concentrated and purified via size exclusion chromatography on a Superdex S200 column (GE Healthcare) equilibrated in buffer (10 mM Tris/HCl pH 7.4, 150 mM NaCl). For VHH-BS1, the eluted protein was concentrated and

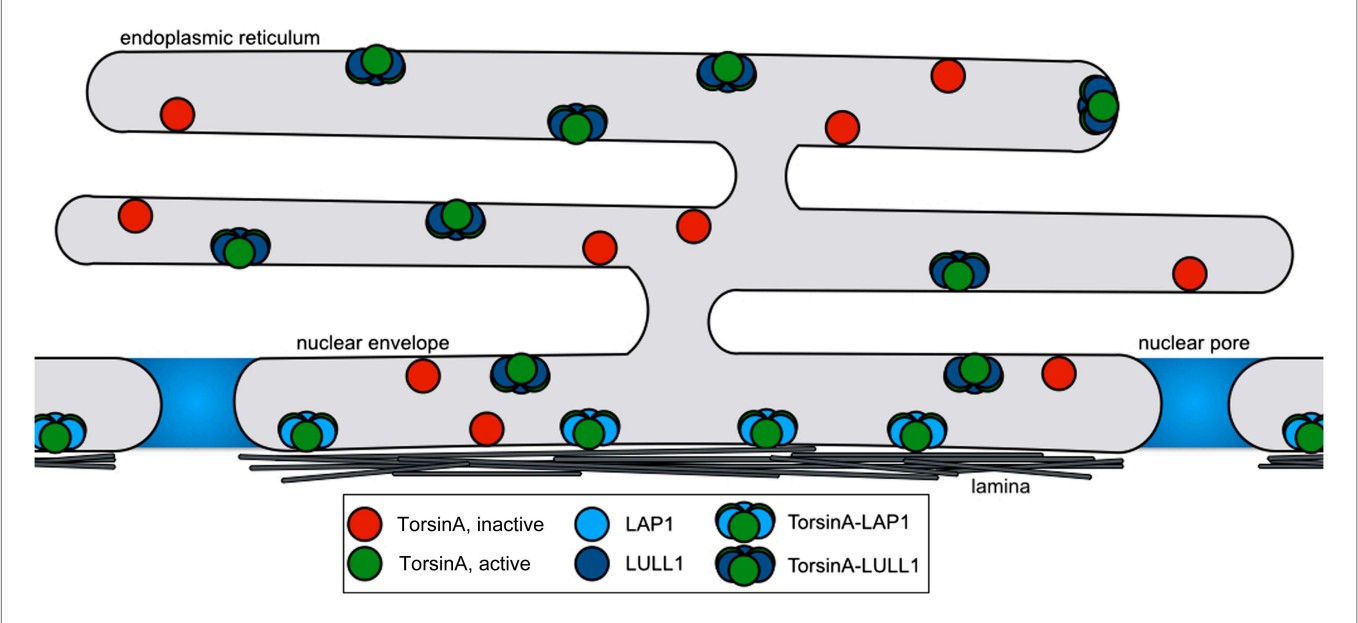

**Figure 5**. Model for Torsin activation and localization. LAP1 (light blue) is localized to the nuclear envelope due to its interaction with the nuclear lamina, while LULL1 (dark blue) is found throughout the endoplasmic reticulum. Both proteins can bind inactive TorsinA (red) and target it to their respective locations. Both LAP1 and LULL1 activate TorsinA (green) when assembled into heterohexameric rings.

purified via size exclusion chromatography on a Superdex S75 column (GE Healthcare) equilibrated in buffer (10 mM Tris/HCl pH 7.4, 150 mM NaCl). TorsinA-containing preparations were performed according to the same procedure with the following modifications: (1) the lysis buffer contained 10 mM $MgCl_2$, 1 mM ATP, and 50 mM HEPES/NaOH pH 8.0 instead of potassium phosphate; (2) the elution buffer contained 10 mM $MgCl_2$, 1 mM ATP, and 10 mM HEPES/NaOH pH 8.0 instead of potassium phosphate; (3) no ion-exchange step was performed; and (4) size exclusion buffer contained 10 mM $MgCl_2$ and 0.5 mM ATP.

The LAP1-VHH-BS1 complex was formed by mixing the individual components using a twofold molar excess of VHH-BS1 and incubating for 30 min on ice, followed by size exclusion chromatography on a Superdex S75 column (GE Healthcare) equilibrated in buffer (10 mM Tris/HCl pH 7.4, 150 mM NaCl).

## Protein crystallization

Purified LAP1-VHH-BS1 was concentrated to 10 mg/ml prior to crystallization. The complex crystallized in 20% (wt/vol) polyethylene glycol (PEG) 3350, 100 mM Bis-Tris pH 5.5, and 200 mM ammonium sulfate by the hanging drop vapor diffusion method in 2 µl drops at 18°C. Crystals grew within 4–12 days with dimensions of 35 µm × 35 µm × 35 µm. Prior to X-ray data collection, crystals were cryoprotected in the reservoir solution supplemented with 30% (wt/vol) PEG 3350 and 15% (vol/vol) glycerol. Data were collected at beamlines 24ID-C/-E at Argonne National Laboratory.

## Structure determination

Data reduction was carried out using HKL2000 software (*Otwinowski and Minor, 1997*); all other software was collectively used through SBGrid (*Morin et al., 2013*). The LAP1-VHH-BS1 structure was solved using molecular replacement. The MRage pipeline procedure from the PHENIX suite (*Adams et al., 2010*) was used for initial search model determination. The Phaser-MR tool was subsequently used for phasing and initial refinement. In the initial map, with phases only provided by the VHH search model, the outlines of the major secondary structure elements of LAP1 were visible. With iterative model building and refinement, the model phases gradually improved and the electron density maps became better defined. The final model was refined against native data extending to 1.6 Å. The data were cut judged by the $CC_{1/2}$ value for the highest resolution shell, and visual inspection of the 2Fo-Fc map (*Figure 1—figure supplement 2*). Model building was carried out with Coot (*Emsley et al., 2010*) and refinement was done with phenix refine from the PHENIX suite.

## Single-particle electron microscopy imaging and data analysis

Ni-affinity purified TorsinA(E171Q)-LAP1(356–583), TorsinA(E171Q)-LULL1(233–470), LAP1(356–583), and LULL1(233–470) (~0.1 mg/ml) were used for negative stain electron microscopy grids. Continuous carbon-film grids were glow-discharged (PELCO easiGlow, Redding, CA) at 15 mA for 30 s. Next, 5 µl specimen solution was loaded immediately onto the grid and blotted after 15 s. The specimen was then stained by uranyl acetate (1% wt/vol) for 10 s, blotted, and dried. Thirty single-particle electron micrographs were recorded using a 2K × 2K CCD camera on an FEI Tecnai Spirit electron microscope at 80 keV, 98,000× nominal magnification (3.63 Å/pixel). For TorsinA(E171Q)-LULL1(233–470), 808 particles were boxed out and subjected to 'direct classification', an unbiased, reference-free, alignment-free classification function in the PARTICLE software package (www.sbgrid.org/software/title/PARTICLE).

## ATPase activity assay

ATP hydrolysis rates of TorsinA:LAP1 and TorsinA:LULL1 complexes were measured by an NADH-coupled assay (*Norby, 1988*). In this assay, each ATP hydrolysis event allows conversion of one molecule of phosphoenolpyruvate into pyruvate by pyruvate kinase. Thereafter, pyruvate is converted to lactate by L-lactate dehydrogenase, which results in oxidation of a single NADH molecule. Loss of NADH over time, which is quantifiably proportional to ATP hydrolysis rates, is monitored by a decrease in absorbance at 340 nm. All of the assays were conducted at room temperature in a buffer containing 10 mM HEPES/NaOH pH 8.0, 150 mM NaCl, and 10 mM $MgCl_2$, and in the presence of 2 mM ATP. Measurements were performed in triplicates using 5 µM of protein complex. Absorbance was measured in 30 µl reaction volume using a 384-well plate reader. Data analysis was performed using Prism.

## In vitro precipitation experiments

TorsinA(E171Q):LAP1 complex was used in precipitation experiments. First, 100 pmol of TorsinA(E171Q):LAP1 complex was mixed with 200 pmol of VHH molecules and incubated at 4°C in the presence of 10 mM HEPES/NaOH pH 8.0, 150 mM NaCl, 10 mM $MgCl_2$, and 0.25 mM ATP. Reactions were centrifuged at 13,000×*g* for 5 min at different time points to separate soluble and insoluble proteins, and the pellets were solubilized in 8 M urea. Equal volumes of samples from soluble and insoluble proteins were analyzed by SDS-PAGE gel electrophoresis.

## Analytical gel filtration

Size exclusion chromatography was performed on a 10/300 Superdex 200 column in 10 mM HEPES/NaOH pH 8.0, 150 mM NaCl, 10 mM $MgCl_2$, and 0.5 mM ATP. Samples (7.5 nmol) of LAP1 and TorsinA(E171Q):LAP1 were loaded in 200 µl injections and peak fractions were analyzed by SDS-PAGE gel electrophoresis. For comparison, 2.5 nmol of LULL1 and TorsinA(E171Q):LULL1 were loaded in 200 µl injections.

## Bioinformatic analysis

Multiple sequence alignments of Torsin and LAP1 were performed using MUSCLE (*Edgar, 2004*) and visualized with Jalview (*Waterhouse et al., 2009*). Since differentiating between LAP1 and LULL1 is difficult based on primary sequence, we refer to the sequences as LAP1 for simplicity. The species nomenclature used for the alignments is as follows: *Homo sapiens* (hs), *Ornithorhynchus anatinus* (oa), *Gallus gallus* (gg), *Takifugu rubripes* (tr), *Danio rerio* (dr), *Branchiostoma floridae* (bf), *Strongylocentrotus purpuratus* (stp), *Ciona savignyi* (cs), *Ciona intestinalis* (ci), *Nematostella vectensis* (nv), *Caenorhabditis elegans* (ce), and *Drosophila melanogaster* (dm).

## Modeling

The structure of human TorsinA was modeled using HHpred in combination with Modeler through the Bioinformatics Toolkit platform (*Biegert et al., 2006*). The D2 domain of ClpA (PDB entry 1R6B; *Xia et al., 2004*) was picked as the closest template structure. Models were generated with another four closely related structures. While there are a few uncertain loops and some discrepancies in modeling the C-terminal domain, the model of the nucleotide binding site is nearly identically in all cases.

The structure of the heterohexameric TorsinA-LAP1 was modeled based on the hexameric D2 ring within the MecA-ClpC assembly (3PXI; *Wang et al., 2011*), by alternately superimposing the LAP1 structure and the TorsinA model onto neighboring ClpC-D2 domains within the ring.

## Immunization of alpaca

An adult male alpaca (*Lama pacos*) was purchased locally, maintained in pasture, and immunized following a protocol authorized by the Tufts University Cummings Veterinary School Institutional Animal Care and Use Committee. Recombinantly expressed LAP1 was used for immunization following a standard protocol (*Maass et al., 2007*). Following the final boost, peripheral blood lymphocytes (PBLs) were harvested from blood as described (*Maass et al., 2007*).

## VHH library generation

Total RNA was isolated from $10^6$ freshly isolated PBLs using an RNeasy Plus Mini Kit (Qiagen, Hilden, Germany), following the manufacturer's guidelines. First strand cDNA synthesis was performed using SuperScript III reverse transcriptase (ThermoFisher Scientific, Waltham MA) and a combination of poly (A) oligo dT, random hexamer or primers specific to alpaca immunoglobulin, AlCH2 and AlCH2.2. Subsequent PCR amplification of VHH sequences and phage library generation followed the procedure described in *Maass et al. (2007)*, including the use of alpaca-specific primers for VHH gene amplification and a phagemid vector adapted for VHH expression as a pIII fusion. Following transformation into TGI cells (Agilent, Santa Clara, CA), the total number of independent clones was estimated to be $4 \times 10^6$/ml. Ninety six clones were selected at random and sequenced to assess library diversity. The resulting phagemid library was stored at −80°C.

## Generation of M13 phage displaying VHH library

A 1 µl sample of the $4 \times 10^6$ library was used to inoculate 100 ml SOC with 10 µg/ml ampicillin. The culture was grown to mid-log phase, and infected with 100 µl $10^{14}$ pfu/ml VCSM13 helper phage. The culture was then incubated for 2 hr at 37°C, and the cells harvested by centrifugation, and resuspended in 100 ml 2YT, 0.1% glucose, 50 µg/ml kanamycin, and 10 µg/ml ampicillin. Cultures were incubated overnight at 30°C, then centrifuged for 20 min at 8000 rpm, and phage precipitated from the resulting supernatant with 1% PEG 6000/500 mM NaCl at 4°C, and resuspended in PBS.

## Selection of VHHs by phage display

A 100 µg sample of LAP1 was labeled via coupling to primary amines with a fivefold molar excess of Chromalink NHS biotin reagent (Solulink, San Diego, CA) for 90 min in 100 mM phosphate buffer pH 7.4, 150 mM NaCl. The reaction was then run through an Amicon 10 kDa MWCO concentrator (EMD Millipore) to remove unreacted biotin. Incorporation of biotin was monitored spectrophotometrically. A 100 µl sample of MyOne Streptavidin T1 Dynabeads (ThermoFisher Scientific) was blocked in PBS/2% BSA for 2 hr at 37°C. Following blocking, 20 µg biotinylated antigen in PBS was added to the beads, and incubated for 30 min at room temperature (RT), with rotation. The beads were then washed 3× in PBS, and 200 µl of $10^{14}$ pfu/ml M13 phage displaying the VHH library was added in PBS/2% BSA for 1 hr at RT. The beads were then washed 15× in PBS/0.1% Tween 20. Phage was eluted by the addition of ER2738 *E. coli* (NEB, Ipswich, MA) for 15 min at 37°C, followed by elution with 200 mM glycine pH 2.2 for 10 min at RT. The glycine elution was neutralized and pooled with ER2738 culture, plated onto 2YT agar plates supplemented with 2% glucose, 5 µg/ml tetracycline, and 10 µg/ml ampicillin, and grown overnight at 37°C.

This enriched library was then used for a second round of panning as described above with the following exceptions: 2 µg of biotinylated antigen was used as bait, and incubated with 2 µl $10^{14}$ pfu/ml M13 phage displaying the VHH library for 15 min at 37°C, followed by longer washes in PBS/0.1% Tween 20.

## ELISA

Following two rounds of phage display panning, 96 colonies were isolated in 96-well round bottom plates and grown to mid-log phase at 37°C in 200 µl 2YT, 10 µg/ml ampicillin, and 5 µg/ml tetracycline, then induced with 3 mM IPTG, and grown overnight at 30°C. Plates were centrifuged at 2500 rpm for 10 min, and 100 µl of supernatant mixed 1:1 with PBS plus 5% non-fat dry milk was then added to an ELISA plate coated with 1 µg/ml antigen. Following multiple washes in PBS plus 1% Tween 20, anti-Etag-HRP antibody (Bethyl) was added at a 1:10,000 dilution in PBS plus 5% non-fat dry milk for 1 hr at RT. The plate was developed with fast kinetic TMB (Sigma-Aldrich), quenched with 1 M HCl and read out at 450 nm (Spectramax, Molecular Devices). ELISA positive clones were sequenced and subcloned into a bacterial expression vector (see above).

## Acknowledgements

We thank Benjamin M Stinson for help with the ATPase assay. The X-ray crystallography work was conducted at the APS NE-CAT beamlines, which are supported by award GM103403 from the National Institute of General Medical Sciences, NIH. Use of the APS is supported by the US Department of Energy, Office of Basic Energy Sciences, under contract no. DE-AC02-06CH11357.

## Additional information

### Competing interests

JZC: Reviewing editor, *eLife*. The other authors declare that no competing interests exist.

### Funding

| Funder | Grant reference number | Author |
|---|---|---|
| Dystonia Medical Research Foundation | | Brian A Sosa, Thomas U Schwartz |
| National Institute of General Medical Sciences | T32GM007287 | Brian A Sosa |

The funders had no role in study design, data collection and interpretation, or the decision to submit the work for publication.

### Author contributions

BAS, Conception and design, Acquisition of data, Analysis and interpretation of data, Drafting or revising the article; FED, Acquisition of data, Analysis and interpretation of data, Drafting or revising the article; JZC, Acquisition of data, Analysis and interpretation of data; JI, Acquisition of data, Drafting or revising the article; HLP, Analysis and interpretation of data, Drafting or revising the article; TUS, Conception and design, Analysis and interpretation of data, Drafting or revising the article

## Additional files

### Major dataset

The following dataset was generated

| Author(s) | Year | Dataset title | Dataset ID and/or URL | Database, license, and accessibility information |
|---|---|---|---|---|
| Sosa BA and Schwartz TU | 2014 | LAP1(aa356-583), H. sapiens, bound to VHH-BS1 | http://www.pdb.org/pdb/explore/explore.do?structureId=4TVS | Publicly available at RCSB Protein Data Bank. |

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
