## [Decision Letter]

Thank you for sending your work entitled “How LAP1 activates and targets Torsin” for consideration at *eLife.* Your article has been favorably evaluated by Randy Schekman (Senior editor) and 3 reviewers, one of whom, Wesley Sundquist, is a member of our Board of Reviewing Editors, and one of whom, James Berger, has agreed to reveal his identity.

The Reviewing editor and the other reviewers discussed their comments before we reached this decision, and the Reviewing editor has assembled the following comments to help you prepare a revised submission.

The manuscript by Sosa et al. describes the structure of the extra-membranal domain of LAP1, a nuclear-envelope associated protein that activates the AAA+ ATPase TorsinA. The function of TorsinA is not yet well defined, but the protein is of considerable interest because mutations lead to movement disorders and because there is increasing evidence that the protein may be a key component of the recently described nuclear envelope budding system. LAP1 and LULL1 are homologous cofactors that activate TorsinA, but their mechanism of action has not been clear. The structure unexpectedly reveals that LAP1 also adopts a cryptic AAA+ ATPase fold, albeit apparently without a functional site for ATP binding and hydrolysis. This suggested the very intriguing possibility that LAP1 and LULL1 activate TorsinA by forming an alternating heterohexameric rings in which the catalytically active TorsinA subunits must pair with inactive LAP1 or LULL1 subunits that donate an “arginine finger” to the TorsinA active sites. The authors present evidence in support of this hypothesis, including: A) Phylogenetic analyses reveal that LAP1 homologs can be found in species that have TorsinA homologs, and that the putative LAP1/LULL1 arginine finger residue is well conserved, B) Class averages from negatively stained EM images of co-expressed and co-purified LULL1-TorsinA complexes show hexameric rings, and C) Mutation of the putative LAP1 arginine finger residue (R563A) reduces the ATPase activity of the LAP1-TorsinA complex by approximately two-fold in a steady-state ATPase activity assay.

The LAP1 structure and TorsinA activation model are of general interest and will advance the field owing to the novelty of the activation mechanism, the apparent biological significance of TorsinA, and the emerging importance of heterologous AAA+ ATPases, particularly in metazoans. This said, the paper as it presently stands is overly abbreviated and incomplete, additional data are needed to more firmly ground some of the claims, (particularly the evidence for heterohexamers), and the manuscript needs extensive editing to remove statements that go beyond the data presented.

Specific issues that must be addressed prior to publication:

1) The evidence that the objects being imaged by EM analyses comprise 1:1 heterohexameric rings of TorsinA(E171):LULL1 is not yet compelling. The authors should: A) show the gels that demonstrate they are purifying 1:1 complexes between TorsinA and both of its activating partners. B) Show raw images of individual assemblies from which the class averages are derived, and C) Perform controls in the activators and TorsinA are left out and demonstrate that hexameric rings are not observed (to rule out the possibility that the rings being imaged are homohexamers of TorsinA(E171) or activators). They should also need to do at least one additional experiment that solidifies their claim of heterohexamer formation. We don't want to dictate exactly what experiment must be done, but have suggested some possibilities in the following section (Point #3).

2) The ATPase assays shown in Figure 3 should be more comprehensive. The LAP1 R563A mutation clearly seems to reduce ATP hydrolysis by TorsinA (although the effects are surprisingly modest, perhaps because the measurement shows steady-state accumulation of hydrolysis products rather than single turnover kinetics?). However, making mutants and testing ATP hydrolysis rates seem straightforward now that the assay is up and running. The authors should therefore at least show data for: A) the TorsinA alone control, B) a set of analogous experiments for the LULL1 activator (which is the protein that they are showing in their EM reconstructions), and C) A more complete analysis of ATPase activity: in addition to showing a bar graph summary of their ATPase data, the authors need to show rates plotted as a function of time and at different ATP concentrations.

3) The manuscript needs to be edited heavily for typographical errors, awkward phrasing, incomplete experimental methods and claims that exceed the data. Specific examples are given in the following section.

Issues for the authors' consideration:

1) Figures. The representations of the structure and its implications are not as clear as they could be. The authors should model the hetero-hexamer formed by a modeled torsinA and the actual LAP1, and also try to match Figure 1 and especially Figure 2 to previous representations of the ASCE domain in AAA ATPases. Additionally, the position of the putative arginine finger is difficult to understand as represented. Consider looking at previous AAA+ ATPase reviews (for example the [32] review) for ideas on how to most clearly present the data and ideas.

2) Nomenclature. Consider a new designation for the structure as it does not include the canonical motifs and C-terminal small domain characteristic of all known AAA+ ATPases.

3) Heterohexamer Analysis. The model would be significantly strengthened by additional biochemical analyses. Can the authors design informative experiments of one or more of the following types? A) Gel filtration analysis to demonstrate that TorsinA(E171) and the activators bind one another and form larger complexes that are consistent with the heterohexamer model, B) Structure-based cross-linking experiments, and/or C) Different tagging of TorsinA and LAP1 to show and quantify heterocomplex formation? Please also consider using these types of experiments to test whether LAP1 or LULL1 arginine-finger mutations block binding and whether complex formation is nucleotide dependent, and whether VHH-BS1 can block Lap1 stimulation of TorsinA ATPase activity in trans (to support the statement, “We note that VHH-BS1 binds to the area immediately surrounding the arginine-finger of LAP1, thereby interfering with TorsinA binding”).

4) Mutational Analyses. The authors should also seriously consider making Cys mutants that begin to test their concept of redox regulation (e.g., is the conserved LAP1 Cys496 residue that they model as being near the enzymatic active site required for activity?).

5) Editing:

A) Title: this manuscript addresses how LAP1 activates torsin, it does not address the question of how LAP1 targets torsin. Title would appropriately read “How LAP1 activates Torsin”, perhaps with some additional information on the suggested mechanism.

B) Very few data are shown for the crystal structure determination process. Figures depicting electron density for a portion of the model, highlighting where the camelid antibody binds (and the contacts formed therein), etc. should be shown. Do the LAP1 domains pack in an interesting/informative manner in the crystal (e.g., through pseudo AAA+ contacts)?

C) Concerning the statement, “While we do not find a conserved arginine in TorsinA at the expected position at the end of helix α5...” This absence should be shown through a multiple sequence alignment.

D) Methods: the description of the EM work is very thin. No raw images of particles are shown. Were the grids glow discharged? How long was staining? Were the grids washed? Also, the 2D class averages look a bit rough and there appear to be both hexameric and heptameric species - comment? Only LULL1-TorsinA particles are shown, but the text implies that LAP1-torsin rings were also seen. These should be shown.

E) Figure 2. It would be helpful to see/show how the LAP1 and torsin AAA+ domains superpose on the template dimer (ClpX?).

F) Table 1. At 1.7 Å, it is surprising that any Ramachandran outliers are present in the model. These could probably be cleaned up.

---

## [Author Response]

*1) The evidence that the objects being imaged by EM analyses comprise 1:1 heterohexameric rings of TorsinA(E171):LULL1 is not yet compelling. The authors should: A) show the gels that demonstrate they are purifying 1:1 complexes between TorsinA and both of its activating partners. B) Show raw images of individual assemblies from which the class averages are derived, and C) Perform controls in the activators and TorsinA are left out and demonstrate that hexameric rings are not observed (to rule out the possibility that the rings being imaged are homohexamers of TorsinA(E171) or activators). They should also need to do at least one additional experiment that solidifies their claim of heterohexamer formation. We don't want to dictate exactly what experiment must be done, but have suggested some possibilities in the following section (Point #3)*.

We have included a gelfiltration experiment (new Figure 3—figure supplement 1), which shows the co-purification of TorsinA(E171Q):LAP1 and TorsinA(E171Q):LULL1 in comparison to LAP1 and LULL1 purified alone. Both complexes elute at a volume consistent with a heterodimeric assembly (theoretical MW is 58.3 kDa), while LAP1 and LULL1 elute at a volume indicative of a monomer. In addition, we have performed EM analysis with LAP1 and LULL1 alone. On the resulting grids, under identical assay conditions as were previously used for the TorsinA(E171Q):LULL1 and TorsinA(E171Q):LAP1 grids, no particles are observed (new Figure 3—figure supplement 2). Both experiments together are consistent with the notion that the hexameric rings observed on TorsinA(E171Q):LULL1 grids are best explained by the formation of trimers built from the stable TorsinA(E171Q):LULL1 heterodimeric units. Since TorsinA without LAP1 or LULL1 present is insoluble in our hands, we do not consider homohexameric TorsinA assemblies to be a possible explanation for our EM objects. To further support the latter point we set up a competition experiment in which we incubated stoichiometric TorsinA(E171Q):LAP1 complex with VHH-BS1 or a control VHH over several hours (new Figure 1—figure supplement 3). VHH-BS1 competes with TorsinA(E171Q) for LAP1 binding and partially displaces TorsinA(E171Q). Free TorsinA(E171Q) then precipitates out of the solution, confirming the solubility problems. With the control VHH, the TorsinA(E171Q):LAP1 complex remains stable.

*2) The ATPase assays shown in*
Figure 3
*should be more comprehensive. The LAP1 R563A mutation clearly seems to reduce ATP hydrolysis by TorsinA (although the effects are surprisingly modest, perhaps because the measurement shows steady-state accumulation of hydrolysis products rather than single turnover kinetics?). However, making mutants and testing ATP hydrolysis rates seem straightforward now that the assay is up and running. The authors should therefore at least show data for: A) the TorsinA alone control, B) a set of analogous experiments for the LULL1 activator (which is the protein that they are showing in their EM reconstructions), and C) A more complete analysis of ATPase activity: in addition to showing a bar graph summary of their ATPase data, the authors need to show rates plotted as a function of time and at different ATP concentrations*.

A control for TorsinA alone is not feasible for the reasons explained in point 1. Analogous data for LULL1 have been included in the revised manuscript and are consistent with the data obtained for LAP1 (The ATPase assay is shown in Figure 4 in the revised manuscript, instead of the original Figure 3). Hydrolysis rate plots have been added to the figure. We have not included data measured at different ATP concentrations. If the reviewers’ concern is product inhibition, the fact that the ATP hydrolysis rate does not change over the course of the experiment is proof that this should not be a concern.

We agree with the notion that the effect of LAP1 R563A and LULL1 R449A on the measured ATP hydrolysis rates is rather small. In this context it is important to note that the rates we measure are obtained in vitro, in the absence of a substrate, and with truncated proteins. We suspect that the situation in vivo, in proximity to the ER membrane and in the presence of substrate, might be drastically different. Therefore, we suggest to simply take away the fact that the LAP1/LULL1 Arg-mutants undoubtedly reduce ATP hydrolysis by Torsin1A, but for a more quantitative analysis the proper TorsinA substrate(s) needs to be identified first.

*3) The manuscript needs to be edited heavily for typographical errors, awkward phrasing, incomplete experimental methods and claims that exceed the data. Specific examples are given in the following section*.

*Issues for the authors' consideration*:

*1) Figures. The representations of the structure and its implications are not as clear as they could be. The authors should model the hetero-hexamer formed by a modeled torsinA and the actual LAP1, and also try to match*
Figure 1
*and especially*
Figure 2
*to previous representations of the ASCE domain in AAA ATPases. Additionally, the position of the putative arginine finger is difficult to understand as represented. Consider looking at previous AAA+ ATPase reviews (for example the*
[32]
*review) for ideas on how to most clearly present the data and ideas*.

The heterohexamer model has been included (new Figure 3). The orientation of LAP1 in Figure 1 is very close to the canonical view of AAA+ ATPases, as it was used, for example, in [7]. For the nucleotide-binding pocket figure (Figure 2), we have tried other orientations, but came to the conclusion that our original orientation still works best for illustrating the major point.

*2) Nomenclature. Consider a new designation for the structure as it does not include the canonical motifs and C-terminal small domain characteristic of all known AAA+ ATPases*.

Agreed. We suggest the name AAA+Activator.

*3) Heterohexamer Analysis. The model would be significantly strengthened by additional biochemical analyses. Can the authors design informative experiments of one or more of the following types? A) Gel filtration analysis to demonstrate that TorsinA(E171) and the activators bind one another and form larger complexes that are consistent with the heterohexamer model, B) Structure-based cross-linking experiments, and/or C) Different tagging of TorsinA and LAP1 to show and quantify heterocomplex formation? Please also consider using these types of experiments to test whether LAP1 or LULL1 arginine-finger mutations block binding and whether complex formation is nucleotide dependent, and whether VHH-*BS1 *can block Lap1 stimulation of TorsinA ATPase activity in trans (to support the statement, “We note that VHH-*BS1 *binds to the area immediately surrounding the arginine-finger of LAP1, thereby interfering with TorsinA binding”)*.

See response to major point 1.

*4) Mutational Analyses. The authors should also seriously consider making Cys mutants that begin to test their concept of redox regulation (e.g., is the conserved LAP1 Cys496 residue that they model as being near the enzymatic active site required for activity?)*.

We agree that these are excellent experiments to do, but they exceed the scope of this manuscript. Also, knowing the substrate(s) would make the suggested analyses much more compelling.

*5) Editing*:

*A) Title: this manuscript addresses how LAP1 activates torsin, it does not address the question of how LAP1 targets torsin. Title would appropriately read “How LAP1 activates Torsin”, perhaps with some additional information on the suggested mechanism*.

We changed the title.

*B) Very few data are shown for the crystal structure determination process. Figures depicting electron density for a portion of the model, highlighting where the camelid antibody binds (and the contacts formed therein), etc. should be shown. Do the LAP1 domains pack in an interesting/informative manner in the crystal (e.g., through pseudo AAA+ contacts)*?

We included a new supplementary figure showing the LAP1-VHH-BS1 complex with electron density (new Figure 1—figure supplement 2). VHH-binding to LAP1 is also shown in comparison to the presumed TorsinA binding site (new Figure 1—figure supplement 3). The Materials and methods section describing the structure determination has been expanded.

*C) Concerning the statement, “While we do not find a conserved arginine in TorsinA at the expected position at the end of helix α5...” This absence should be shown through a multiple sequence alignment*.

We included the MSA (new Figure 2—figure supplement 1).

*D) Methods: the description of the EM work is very thin. No raw images of particles are shown. Were the grids glow discharged? How long was staining? Were the grids washed? Also, the 2D class averages look a bit rough and there appear to be both hexameric and heptameric species - comment? Only LULL1-TorsinA particles are shown, but the text implies that LAP1-torsin rings were also seen. These should be shown*.

Raw images have now been included (Figure 3—figure supplement 2). The Methods section has been extended. Regarding the ambiguous heptameric appearance of a small fraction of the 2D class averages, we speculate that these might be the result of slightly tilted side views and/or somewhat flexible conformations. To provide an ultimately conclusive answer requires a much larger dataset suitable for high-resolution structural analysis, or, even better, a crystal structure.

*E)*
Figure 2*. It would be helpful to see/show how the LAP1 and torsin AAA+ domains superpose on the template dimer (ClpX?)*.

The new Figure 3 now includes a heterohexameric model of LAP1 and TorsinA superposed on the ClpC-D2 homohexameric ring.

*F)*
Table 1*. At 1.7 Å, it is surprising that any Ramachandran outliers are present in the model. These could probably be cleaned up*.

We have found a better crystal, which belongs to space group P21 and diffracted to 1.6Å. The resolution cut was based on the CC1/2 value and visual inspection of the 2Fo-Fc electron density map. The P21 dataset has generally better data collection and refinement statistics, thus we replaced the original P1 dataset with the new P21 dataset in the revised manuscript. The LAP1-VHH-BS1 model built in both space groups is virtually identical.